# PRO-based Stratification Improves Model Prediction for Toxicity and Survival of Head and Neck Cancer Patients

Eric A. Anyimadu[1]    Yaohua Wang[1]    Carla Floricel[2]    Serageldin Kamel[3]    Clifton David Fuller[3]

Amy Catherine Moreno[3]    Xinhua Zhang[2]    G. Elisabeta Marai[2]    Guadalupe Canahuate[1]

*Abstract*—**Patient-Reported Outcomes (PRO) consist of information provided directly by the patients about their health status including symptom ratings. PROs are commonly used in clinical practice to support clinical decision-making and have recently been incorporated into machine learning models to improve risk prediction. In this work, we aim to evaluate whether the inclusion of a patient stratification based on 12-month post-treatment predicted Patient Reported Outcomes improves risk prediction of radiation-induced toxicity and overall survival for head and neck cancer patients. A bidirectional long-short term memory (Bi-LSTM) recurrent neural network was used to model the longitudinal PRO data and to predict symptom ratings 12 months post-treatment. Patients were stratified using hierarchical clustering over the LSTM-predicted data. A logistic regression model was trained to predict Xerostomia at 12 months and a Cox regression model to predict overall survival. Results show that the inclusion of symptom burden clusters derived from the predicted Patient Reported Outcomes improves radiation-induced toxicity and overall survival prediction for head and neck cancer patients.**

*Index Terms*—**Patient Reported Outcomes, Deep Learning, Patient Clustering, Regression, Survival Analysis, Xerostomia**

## I. INTRODUCTION

Personalized therapeutics in oncology have resulted in a greater variety of head and neck cancer (HNC) treatment outcomes for patients. Despite the increase in survival outcomes, in many patients, treatment leads to long-lasting or permanent residual sequelae [1], whose severity, rate of development, and resolution after treatment vary largely between survivors [2]–[6]. One of the confounders routinely encountered with models of radiation sequelae is that while many patients experience acute side effects, there is interval recovery, limiting the predictive capacity of simple dosimetric or clinic dosimetric models over extended post-treatment time intervals. For example, while the majority of patients experience enhanced xerostomia on-treatment, a minority have moderate-severe xerostomia by 12 months on Phase III randomized trials of intensity-modulated radiation therapy (IMRT) [8].

At the same time, patient-reported outcomes extracted from questionnaires [9] offer important information that can improve clinical decision-making and individual care delivery [10] and could be critical for the efficient prediction of symptoms and survival outcomes in patients. The M.D. Anderson Cancer Center documents and quantifies head and neck cancer symptoms through a standardized monitoring program based on the M.D. Anderson Symptom Inventory (MDASI) [11], a patient-reported outcome (PRO) measure for clinical and research use. The program uses questionnaires that are collected weekly at the time of the treatment appointment, at the acute stage, and at longer intervals post-treatment late stage, during cancer recurrence monitoring. However, in HNC, predicting symptom risk is particularly challenging due to the combined effects of specific treatments and clinical factors [12]. Additionally, some symptoms are correlated, either through direct influence or shared underlying causes. These factors hamper personalized care and make predicting treatment outcomes difficult. As a result, patient clusters based on symptom burden can be leveraged to understand how symptoms are correlated with the diagnosis, clinical attributes, and prescribed treatment [7], [13]–[18].

In this work, we use a Bidirectional Long Short Term Memory (Bi-LSTM) recurrent neural network to model the PRO data [19], [20]. By iteratively applying Bi-LSTM models at each time point, we are able to predict long-term symptom ratings (i.e. 12 months after treatment) starting from the baseline. We then apply clustering to the predicted ratings to identify three symptom burden clusters (i.e. low, medium, high). The cluster labels are then used in the prediction of toxicity and survival for head and neck cancer patients. Since symptom trajectory is predicted using only baseline information, or information that is available at diagnosis or shortly after diagnosis, the predictive models including symptom cluster burden can be applied before treatment starts.

Our results show that the inclusion of symptom-burden clusters when predicting Xerostomia at 12 months improved the test AUC of a logistic regression model from 0.54 to

This work was partially supported by NIH award NCI-R01-CA258827.

[1]E. Anyimadu, [1]Y. Wang, and [1]G. Canahuate are with the University of Iowa, Iowa City, IA 52242 USA (e-mail: eric-anyimadu, yaohua-wang, guadalupe-canahuate@uiowa.edu).

[2]C. Floricel, [2]X. Zhang, and [2]G. Marai are with the University of Illinois Chicago, Chicago, IL 60607 USA (e-mail: cflori3, zhangx, gmarai@uic.edu).

[3]S. Kamel, [3]A. Moreno, and [3]C. Fuller are with the University of Texas MD Anderson Cancer Center, Houston, TX 77030 USA (e-mail: skattia, akmoreno, dfuller@mdanderson.org).

0.86. For this model, the high-symptom cluster label was the predictor with the highest odds ratio (3.132; [1.832, 5.355]). Moreover, the inclusion of the cluster labels in the survival Cox model improved the test concordance index from 0.62 to 0.67. The largest hazard ratio is reported for cancer staging (T and N stages) followed by the high symptom burden cluster.

The contributions of this paper can be summarized as:

- Apply Bi-LSTM for prediction of 12-month symptom burden from baseline ratings
- Cluster the patients using the Bi-LSTM predicted ratings into low-, mid-, and high-symptom burden clusters
- Show that inclusion of the predicted symptom burden significantly improves the predictive performance for Xerostomia 12-month after treatment
- Show that the symptom burden clusters also improve survival prediction when included in a Cox Model

## II. RELATED WORK

PRO data, which capture patients' self-assessments of their health, have been increasingly integrated in recent years into statistical models to improve personalized treatment strategies [26], [29] and used to characterize the symptom burden experienced after treatment [28]. The utility of PRO data in identifying adverse events in the quality of life and enhancing decision-making in cancer treatment management, particularly when coupled with traditional clinical indicators has also been established [25], [27]. However, incorporating PRO into predictive models used in clinical practice remains elusive with missing data being one of the biggest challenges [30].

Recently, machine learning techniques have been applied successfully to the imputation of PRO data outperforming other more traditional methods [19], [31]. Furthermore, Long Short-Term Memory (LSTM) models have been shown to be effective for predicting long-term post-treatment symptom severity in head and neck cancer patients [20].

While clustering has been applied to PRO data to identify symptom clusters and characterize the heterogeneity of symptom burden for cancer patients [6], [9], predictive models including PRO data have mainly used individual symptom ratings [5] and have not leveraged patient stratification based on symptom rating trajectories. To address this gap, we propose an approach that applies patient stratification to predict both toxicity and overall survival in head and neck cancer patients. Our study proposes a novel framework that integrates hierarchical clustering of Bi-LSTM-predicted PRO data with clinical data to improve the performance of traditional regression models. This combined approach aims to enhance the accuracy of radiation-induced toxicity and survival predictions, ultimately providing more personalized and clinically meaningful insights.

## III. METHODS AND MATERIALS

### A. Data

Data was collected from a cohort of 937 head and neck cancer patients from the MD Anderson Cancer Center in Texas who were treated using radiation therapy (RT) between 2010

TABLE I
LIST OF HEAD AND NECK CANCER-SPECIFIC SYMPTOMS AND GENERAL CANCER SYMPTOMS FROM THE MDASI-HN QUESTIONNAIRE.

| Symptom | Category |
|---|---|
| swallow, speech, mucus, taste, constipation, teeth, sores, choking, skin | HNC cancer |
| fatigue, sleep, distress, pain, drowsiness, sadness, memory, numbness, dry mouth, appetite, shortness of breath (sob), nausea, vomit | General cancer |

and 2021. The patient data, extracted from medical records, include clinical and treatment information, and patient-reported symptom ratings. The clinical attributes used in this work include demographics: age, gender, and smoking status; diagnostic attributes include tumor size, lymph node stage, and tumor sub-site. Treatment attributes include indicators as to whether the patient received induction therapy (IC), concurrent therapy (CC), and/or neck dissection surgery (ND). All patients underwent radiation therapy.

Symptom burden data are collected using patient-reported outcome (PRO) questionnaires based on MDASI-HN (MD Anderson Symptom Inventory, the Head and Neck Module) [11], a 28 symptom inventory. In the questionnaire, patients are asked to rate symptoms using a 0-to-10 scale, from "not present" (0) to "as bad as you can imagine" (10). Symptoms are grouped into 3 categories: HNC-specific, general cancer, and six interference symptoms. In this work, we focus on the 22 HNC-specific and general cancer symptoms, listed as a reference in Table I. PRO data are collected prior to treatment and subsequently at multiple points during and after the treatment process. During treatment, a spike in symptom burden is expected due to treatment toxicity with most symptoms subsiding over time. However, for some patients, toxicity treatment leads to long-lasting sequelae. Dry mouth and taste are some of the most prevalent symptoms for oropharyngeal cancer patients.

As we are interested in evaluating the predictive performance of late symptom burden in toxicity and survival, we include the PRO symptom data available before treatment (i.e. baseline, denoted as B), at the end of treatment (W0), and during the post-treatment observation period, which includes 6 weeks (W6), 6 months (M6), and 12 months (M12) after treatment.

This retrospective study was exempt under MD Anderson IRB protocol RCR-003-0800. In compliance with the Health Insurance Portability and Accountability Act (HIPAA), informed consent was waived and approved by the IRB as all analyses were performed over retrospective anonymized data.

### B. Bi-LSTM Model

We model the longitudinal PRO symptom data using a bidirectional long short-term memory (Bi-LSTM) model [19]. Bi-LSTM neural networks contain two LSTM layers that learn information by training using both the forward and backward directions of the PRO longitudinal data. Compared to the

traditional LSTM model, Bi-LSTM can capture additional upstream information by concatenating the hidden states from both LSTM layers and making better predictions.

We applied Bi-LSTM models for both missing data imputation and M12 prediction. For patients with missing baseline (B) ratings, we used cohort mean values to initialize the LSTM. After imputing baseline ratings, we applied the Bi-LSTM iteratively to predict subsequent time points (W0, W6, M6, and M12). We used 3-fold cross-validation for each symptom, training on two folds and testing on the third, ensuring each patient was in the test fold once. Predictions from the test fold were used to identify symptom burden clusters for each patient.

The Bi-LSTM model used in this work consists of one layer of Bi-LSTMs with 10 units followed by a dense layer for the prediction. The input to the model is a sequence where the length is the number of patients and the feature size is 22 per time point. The output is a 22-dimension vector representing the predicted ratings for 22 symptoms at the next time point. We trained the Bi-LSTM model at each time point with internal validation using a 70/30 train/test split over the two train-fold for each iteration of the three-fold cross-validation. Each model was trained with the SGD optimizer using 0.215 as the learning rate with early stopping and 1000 epochs. The mean squared error (MSE) loss function was used to help find the optimum. The implementation used the TensorFlow package and training was done using an NVIDIA RTX 4080 GPU.

By quantifying the computational workload, we provide both the theoretical and empirical costs for training and testing of the Bi-LSTM model. Assuming each gate takes one floating point operation (FLOP), and given that the Bi-LSTM model has 22 features, 10 hidden dimensions, and 10 LSTM units, a single pass through one LSTM unit would need 1280 FLOP and the forward and backward passes over the 10 LSTM units for 4 time steps would take $3.07 \times 10^5$ FLOP. Since we are using 3-fold cross-validation and training for 1000 epochs, the total computational workload is about $1.52 \times 10^{12}$ FLOP. Using the RTX 4080 platform which can handle 48 TFLOP/sec, the theoretical time for model training is $3.17 \times 10^{-2}$ seconds. Empirically, the average training time per time step was roughly 4.2 seconds with a total training time around 17 seconds. The difference in performance can be attributed to the overheads associated with the use of the Spyder IDE on the Anaconda platform and the Tensorflow Package, the data transfer between CPU and GPU, the I/O, Python, and interpreter overheads, and so on. The measured testing time was only 0.02 seconds.

### C. Patient Stratification

To determine patient stratification and associations in terms of treatment-related toxicity, we employed a hierarchical clustering technique on the PRO for 22 symptom Bi-LSTM-predicted ratings at a specific time point. Hierarchical clustering is an unsupervised learning method that builds a hierarchy of clusters by progressively merging or splitting existing

clusters based on similarity measures. We used Euclidean distance as the measure of similarity between the symptom ratings of two different patients.

The Ward method was utilized as the linkage function. This method aims to minimize the total within-cluster variance at each step of the clustering process. Essentially, the Ward method merges clusters in a way that produces the smallest possible increase in the sum of squared differences within each cluster. This approach tends to create clusters of relatively proportional size and variance, enhancing the interpretability and coherence of the resulting patient stratification [21].

We consider patient clusters at baseline, 6 weeks, and 12 months post-treatment. The baseline clusters do not use Bi-LSTM predictions, while the 6 weeks and 12 months are all predicted symptom ratings using the Bi-LSTM. For each time point, we identified three clusters labeled as low, mild, or high, to represent symptom burden level. These labels align with those used in other works [22].

### D. Prediction Models

We considered two different models for toxicity and survival prediction.

*1) Toxicity Prediction:* To evaluate the effectiveness of symptom burden clusters in predicting the development of Xerostomia 12 months post-treatment, we employed a logistic regression model. Xerostomia was coded as a binary outcome, defined by a dry mouth rating of 5 or greater 12 months after treatment. Predictors for this model included age, gender, T and N staging (AJCC $8^{th}$ edition), smoking status, tumor site, and treatment variables (induction/concurrent chemotherapy and neck dissection surgery). We excluded from the analyses near-zero variance attributes where the vast majority of patients belonged to a single category ($> 99\%$). For this reason, M-staging and HPV (p16) status were excluded, as most patients in the cohort are M stage 0 and HPV positive.

All attributes were treated as categorical attributes. Age was categorized into two groups: less than 65 and 65 and over. For attributes where categories did not meet a minimum threshold for support (e.g. at least 10% of patients), categories were sensibly merged with other categories to ensure statistical robustness. For example, the T stage was simplified into early (T0-T2) and late (T3, T4) stages and similarly, the N stage was also consolidated into two categories: (N0, N1) and (N2, N3). For the tumor sub-site, a new category "Other" was created to represent any sub-site other than Base of Tongue (BOT) and Tonsil, which were well represented in the data.

Each predictor was processed using one-hot encoding to convert them into binary representations. To facilitate model interpretation, the most common category, aside from the symptom burden labels, was used as the reference. For symptom burden cluster labels, the mild label served as the reference category.

All features and their categories used in the prediction models are detailed in Table II.

*2) Survival Analysis:* A Cox proportional hazard regression model (Cox model) was used for overall survival analysis.

We used the survival/follow-up time in months and the event flag (dead/alive) as the censored outcome for the Cox model. The same predictors used for toxicity were used for survival analysis.

### E. Evaluation

In addition to the RMSE, to evaluate the Bi-LSTM performance, we also consider three different symptom rating thresholds. Threshold 1 (rating $\geq 1$) evaluates performance for symptom occurrence. Thresholds 3 and 5 (rating $\geq 3$ and rating $\geq 5$) are considered mild and moderate to severe symptom ratings respectively. Using a binary indicator for each symptom as an outcome, allows us to evaluate the Bi-LSTM performance using AUC and confusion matrices.

To evaluate the predictive performance of the Logistic and Cox models, we compare four models. The model without symptom burden clusters served as the base model. We also considered three additional models that incorporate a symptom burden cluster from baseline (B), week 6 (W6), and 12 months (M12) post-treatment. These four logistic regression/Cox models are referenced with the suffixes: wo-cluster, w-cluster-B, w-cluster-W6, and w-cluster-M12, respectively. The logistic and Cox models were non-penalized, meaning no regularization was applied to the coefficients learned by the models.

The evaluation metrics for the logistic regression models included the area under the ROC curve (AUC) [23], and for the Cox models, the concordance index (C-index) [24]. For validation of the predictive models, we used an 80/20 train-test split, repeated ten times. We report the average metric for all ten runs along with the standard deviation. Performance metrics were reported for both the training and testing sets. Additionally, we included the odds ratio (OR) for each model.

## IV. RESULTS

### A. Data

Table II provides the distribution of patients and clinical attributes for the entire cohort and each of the symptom burden clusters. A total of 937 patients were included in the analysis. Out of this, a majority (66.4%) were less than 65 years old and 90.6% were also male. Non-smokers constituted 56.3% of the cohort while the majority had tumor(s) located in the tonsil or base of the tongue. Also, most of the patients were diagnosed in the early T and N stages. In terms of treatment, most of the patients received concurrent chemotherapy (72.20%), 18.5% received induction chemotherapy, and 15.4% received neck dissection surgery.

Table III shows the average severity (avg_sev) and the percentages of the missing data (%_miss) for each symptom longitudinally. On average, ratings at W6 and M12 have the highest missing rate (44.6% and 43.8% respectively) while the baseline (B) has the lowest missing rate (15.9%). The missing rates for each symptom are fairly consistent, indicating that patients tend to fill all questions in the MDASI-HN questionnaire and when a questionnaire is missed, all symptom ratings will be missing for that time point. As can be seen in Table III, starting from baseline (B), the severity of all

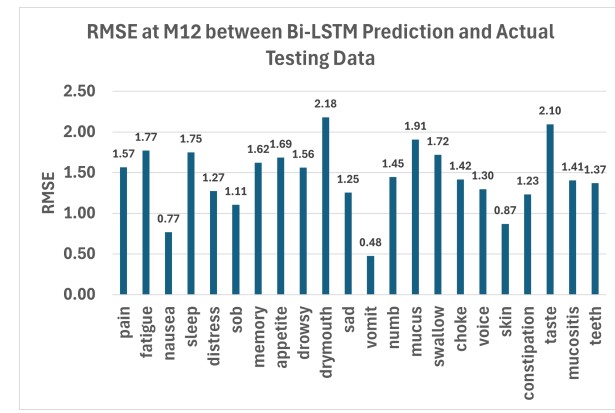

Fig. 1. RMSEs at M12 for all 22 symptoms between Bi-LSTM prediction and actual testing data.

symptoms increases during treatment (W0) and subsides over time for most patients. The most prevalent symptoms for HNC patients are dry mouth and taste with long-term moderate to high severity for a substantial proportion of patients. The symptom with the lowest average rating is vomit followed by nausea.

### B. Bi-LSTM Model

Figure 1 shows the RMSE performance of the Bi-LSTM model for predicting M12 ratings for all 22 general and HNC-specific symptoms. As can be seen, among all the symptoms, vomit achieves the lowest RMSE at 0.48 whereas dry mouth achieves the highest RMSE at 2.18. Not surprisingly, these correspond to the symptoms with the lowest and highest averages at M12. The RMSE for all symptoms is below 2, with the exception of drymouth and taste at 2.18 and 2.10 RMSE, respectively.

For dry mouth and taste symptoms, we evaluate the AUCs and confusion matrices at different rating thresholds and present the results for ratings $\geq 3$ in Figure 2. We chose 3 because it had the most even distribution of patients. At threshold 1, most patients experienced the symptom and at threshold 5, most patients did not. As is shown in the figure, for both symptoms, Bi-LSTM models achieve good AUCs, 0.81 for dry mouth and 0.82 for taste. The confusion matrices also indicate good true positive rate and true negative rate.

### C. Patient Stratification

Three symptom burden clusters were identified from the LSTM predicted 12-month post-treatment (M12) symptom ratings. The majority of the patients (52.9%) are in the low symptom burden cluster, 35.1% in the mild symptom cluster, and 12.0% in the high symptom burden cluster. As can be seen in Table II, the distribution of the clinical covariates within the symptom burden clusters follows the same distribution as the entire cohort. T-stage and neck dissection are significantly associated with the symptom clusters, the toxicity and survival outcomes.

TABLE II

PATIENT DISTRIBUTION FOR THE PREDICTORS AND OUTCOMES USED IN THE PREDICTION MODELS. DISTRIBUTION IS ALSO SHOWN FOR THE 12-MONTH LSTM-PREDICTED PATIENT STRATIFICATION.

| Attribute | Category | All Patients (N=937, 100%) | | Low Symptom Cluster (N=496, 52.9%) | | Mild Symptom Cluster (N=329, 35.1%) | | High Symptom Cluster (N=112, 12.0%) | | P-values |
|---|---|---|---|---|---|---|---|---|---|---|
| Age | <65 | 622 | 66.4% | 338 | 68.1% | 216 | 65.7% | 68 | 60.7% | 0.304 |
| | >=65 | 315 | 33.6% | 158 | 31.9% | 113 | 34.3% | 44 | 39.3% | |
| Gender | male | 849 | 90.6% | 461 | 92.9% | 290 | 88.1% | 98 | 87.5% | 0.033 |
| | female | 88 | 9.4% | 35 | 7.1% | 39 | 11.9% | 14 | 12.5% | |
| Smoking | smoker | 409 | 43.6% | 204 | 41.1% | 146 | 44.4% | 59 | 52.7% | 0.080 |
| | non-smoker | 528 | 56.4% | 292 | 58.9% | 183 | 55.6% | 53 | 47.3% | |
| Site of tumor | BOT | 436 | 46.5% | 210 | 42.3% | 166 | 50.5% | 60 | 53.6% | |
| | Tonsil | 426 | 45.5% | 248 | 50.0% | 133 | 40.4% | 45 | 40.2% | 0.041 |
| | other | 75 | 8.0% | 38 | 7.7% | 30 | 9.1% | 7 | 6.3% | |
| T stage | T0-T2 | 697 | 74.4% | 404 | 81.5% | 224 | 68.1% | 69 | 61.6% | <0.001 |
| | T3,T4 | 240 | 25.6% | 92 | 18.5% | 105 | 31.9% | 43 | 38.4% | |
| N stage | N0,N1 | 689 | 73.5% | 380 | 76.6% | 236 | 71.7% | 73 | 65.2% | 0.030 |
| | N2,N3 | 248 | 26.5% | 116 | 23.4% | 93 | 28.3% | 39 | 34.8% | |
| CC | yes | 677 | 72.3% | 340 | 68.5% | 250 | 76.0% | 87 | 77.7% | 0.026 |
| | no | 260 | 27.7% | 156 | 31.5% | 79 | 24.0% | 25 | 22.3% | |
| IC | yes | 174 | 18.6% | 81 | 16.3% | 64 | 19.5% | 29 | 25.9% | 0.055 |
| | no | 763 | 81.4% | 415 | 83.7% | 265 | 80.5% | 83 | 74.1% | |
| Neck dissection | yes | 145 | 15.5% | 99 | 20.0% | 32 | 9.7% | 14 | 12.5% | <0.001 |
| | no | 792 | 84.5% | 397 | 80.0% | 297 | 90.3% | 98 | 87.5% | |
| **Outcomes** | | | | | | | | | | |
| Xerostomia at 12 months | yes | 207 | 22.1% | 11 | 2.2% | 119 | 36.2% | 77 | 68.8% | <0.001 |
| | no | 730 | 77.9% | 485 | 97.8% | 210 | 63.8% | 35 | 31.3% | |
| Overall Survival (in months) | Median (25% - 75%) | 27, | (16 - 53) | 28, | (18 - 57) | 25, | (15 - 50) | 25, | (14 - 43) | - |
| | dead | 80 | 8.5% | 26 | 5.2% | 34 | 10.3% | 20 | 17.9% | <0.001 |
| | alive | 857 | 91.5% | 470 | 94.8% | 295 | 89.7% | 92 | 82.1% | |

TABLE III

AVERAGE SEVERITY AND PERCENTAGE OF MISSINGNESS AT EACH TIME POINT FOR THE ORIGINAL PROS FOR ALL 22 GENERAL CANCER SYMPTOMS AND HNC CANCER SYMPTOMS.

| Symptom | B | | W0 | | W6 | | M6 | | M12 | |
|---|---|---|---|---|---|---|---|---|---|---|
| | avg_sev | %_miss | avg_sev | %_miss | avg_sev | %_miss | avg_sev | %_miss | avg_sev | %_miss |
| pain | 1.73 | 15.7% | 5.39 | 36.5% | 2.12 | 44.5% | 1.16 | 31.3% | 0.88 | 43.4% |
| fatigue | 1.89 | 15.8% | 4.71 | 36.8% | 2.90 | 44.7% | 2.11 | 31.2% | 1.58 | 43.6% |
| nausea | 0.34 | 15.9% | 2.61 | 36.5% | 0.46 | 44.6% | 0.29 | 31.4% | 0.15 | 43.9% |
| sleep | 1.88 | 15.8% | 3.51 | 36.5% | 2.08 | 44.4% | 1.66 | 31.4% | 1.26 | 43.5% |
| distress | 1.79 | 15.7% | 1.83 | 36.4% | 1.22 | 44.5% | 0.85 | 31.3% | 0.63 | 44.0% |
| sob | 0.55 | 15.7% | 0.88 | 36.4% | 0.69 | 44.5% | 0.48 | 31.4% | 0.48 | 43.5% |
| memory | 0.77 | 15.8% | 1.44 | 36.5% | 1.11 | 44.7% | 1.16 | 31.5% | 1.33 | 43.9% |
| appetite | 0.88 | 15.7% | 4.98 | 36.6% | 2.38 | 44.5% | 1.90 | 31.1% | 0.97 | 43.6% |
| drowsy | 1.29 | 16.1% | 3.78 | 36.6% | 1.96 | 44.5% | 1.60 | 31.4% | 1.13 | 43.9% |
| dry mouth | 0.95 | 15.6% | 5.02 | 36.5% | 4.18 | 44.5% | 4.38 | 31.3% | 3.37 | 43.4% |
| sad | 1.21 | 15.6% | 1.43 | 36.5% | 1.03 | 44.7% | 0.70 | 31.3% | 0.61 | 43.6% |
| vomit | 0.12 | 15.8% | 1.31 | 36.5% | 0.27 | 44.6% | 0.11 | 31.3% | 0.06 | 43.8% |
| numb | 0.50 | 15.9% | 0.96 | 37.0% | 0.68 | 44.7% | 0.90 | 31.5% | 0.85 | 44.2% |
| mucus | 0.98 | 16.1% | 5.64 | 36.6% | 2.78 | 44.6% | 2.36 | 31.3% | 1.67 | 43.9% |
| swallow | 1.19 | 16.4% | 5.06 | 36.6% | 2.58 | 45.1% | 2.18 | 31.4% | 1.80 | 43.8% |
| choke | 0.64 | 16.1% | 2.45 | 36.9% | 0.98 | 44.8% | 1.14 | 31.4% | 1.00 | 43.8% |
| voice | 0.72 | 16.0% | 2.59 | 36.7% | 1.19 | 44.6% | 1.10 | 31.4% | 0.83 | 43.6% |
| skin | 0.21 | 16.2% | 4.06 | 36.9% | 0.61 | 44.4% | 0.29 | 31.3% | 0.24 | 43.6% |
| constipation | 0.63 | 16.1% | 2.79 | 36.7% | 1.20 | 44.6% | 0.81 | 31.2% | 0.54 | 44.0% |
| taste | 0.57 | 16.3% | 6.79 | 36.8% | 4.33 | 44.8% | 3.57 | 31.3% | 2.59 | 43.8% |
| mucositis | 0.64 | 16.1% | 5.16 | 37.1% | 2.31 | 44.8% | 1.06 | 31.5% | 0.52 | 43.9% |
| teeth | 0.48 | 16.4% | 2.34 | 37.1% | 0.97 | 44.7% | 0.83 | 31.5% | 0.70 | 44.0% |
| **Average** | **0.91** | **15.9%** | **3.40** | **36.7%** | **1.73** | **44.6%** | **1.39** | **31.4%** | **1.05** | **43.8%** |

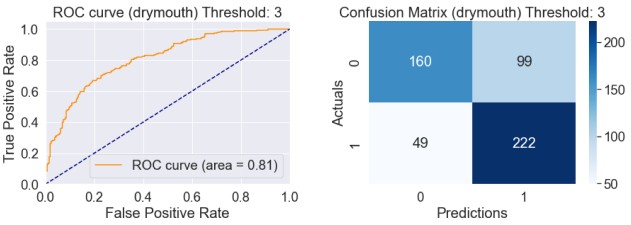

(a) Dry mouth symptom

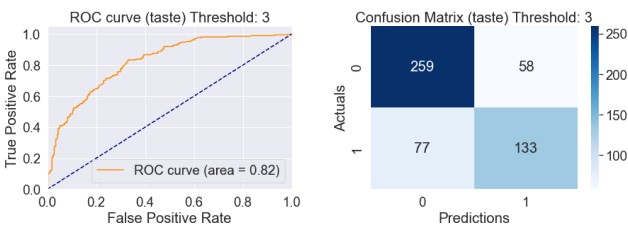

(b) Taste symptom

Fig. 2.  AUCs and Confusion Matrices for 2a Dry mouth symptom and 2b Taste symptom at threshold 3 (moderate).

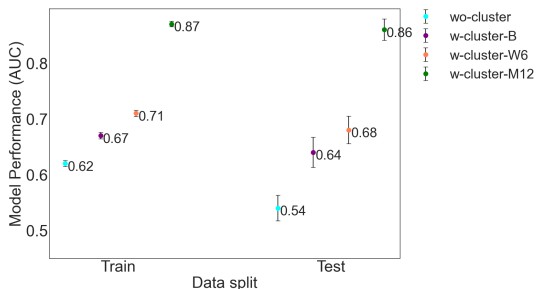

Fig. 3.  Logistic regression model performance over the Train and Test data splits

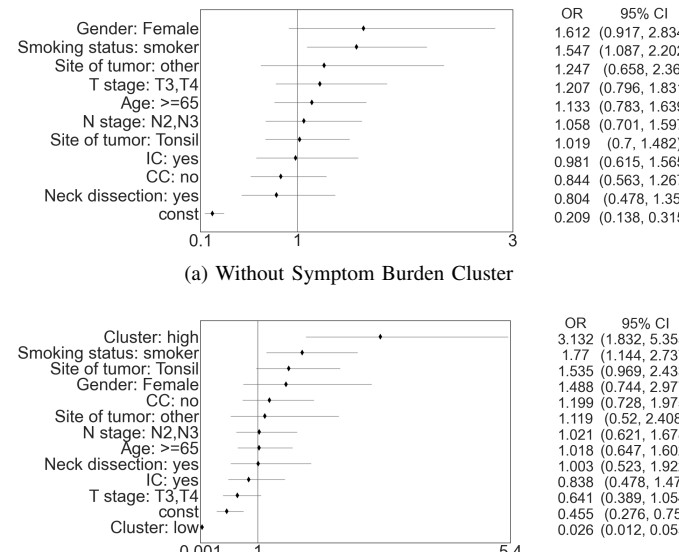

Fig. 4.  Odds ratio for the logistic regression model predictors when (a) the cluster labels are excluded from the model versus (b) when the cluster labels are included as predictors in the model.

Comparing the M12 symptom burden clusters with the same clusters using the baseline (B) and 6 weeks (W6) post-treatment ratings, at baseline, a clear majority (80.6%) of the patients are in the low symptom cluster, and only 2.6% are in the high symptom cluster. For W6, 73.0% of the patients are placed in the low cluster, 22.7% in the mild cluster, and 4.3% in the high cluster.

Figure 5 shows the symptom burden trajectories for the three M12 symptom burden clusters for all symptoms presented into three groups from the least to the most severe symptoms. As can be seen, the high symptom cluster shows average severity for all symptoms when compared to the other two clusters, and the differences are more evident for the most prevalent symptoms (dry mouth and taste). Moreover, even when only M12 were used for clustering, the symptom rating trajectories conform to the severity of the cluster labels.

### D. Toxicity Prediction

Figure 3 shows the average AUC and standard deviation of the logistic regression models over the training and testing data

for the four models evaluated. Overall, the worse performance is observed when no cluster label is included as a predictor of the model, logit-wo-cluster. The best-performing model is the logit-w-cluster-M12 for both the training and test sets. The comparable performance between training and testing indicates that the models are not overfitting.

Figures 4a and 4b illustrate the odds ratios of predictors used in the logistic regression models excluding and including cluster labels (M12) as predictors respectively. In the model excluding cluster labels, the highest odds ratios are observed for female gender and smoking, with smoking being the only significant association with xerostomia. In contrast, when cluster labels are included as predictors, high symptom burden cluster and smoking emerge as the most significant predictors and are significantly associated with xerostomia.

### E. Survival Prediction

Figure 6 shows the Cox models' performance comparison by computing c-index over the train and test data. For both the training and test sets, the cox-wo-cluster, which excludes any cluster labels as predictors, had the lowest performance. The inclusion of the cluster labels into the cox models improved the c-index for both training and testing with cox-w-cluster-W6 showing the best c-index for the train set and the cox-w-cluster-B the best performance over the test set.

Figures 7a and 7b present the Cox models' odds ratios of predictors, both excluding and including the cluster labels (M12) as predictors, respectively. In both models, the predictors significantly associated with survival include advanced T-stage (T3, T4) and N-stage (N2, N3) diagnosis. This association remains even for the model that includes cluster labels.

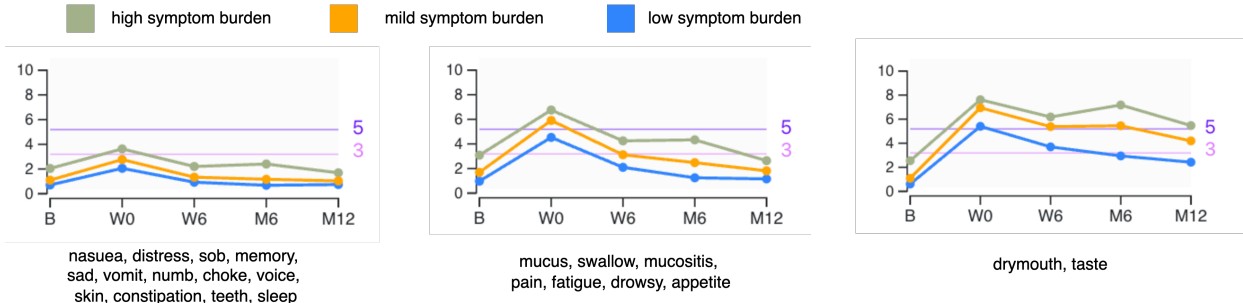

nasuea, distress, sob, memory,
sad, vomit, numb, choke, voice,
skin, constipation, teeth, sleep

mucus, swallow, mucositis,
pain, fatigue, drowsy, appetite

drymouth, taste

Fig. 5. Symptom burden trajectories

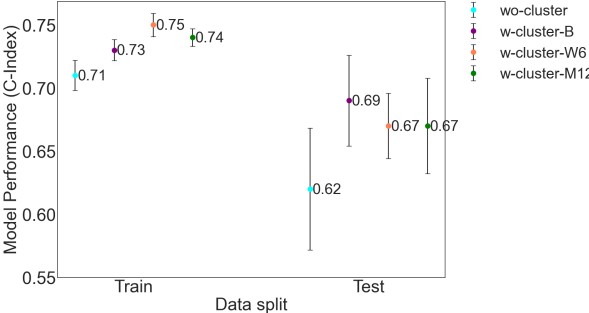

Fig. 6. Cox model performance over the Train and Test data splits

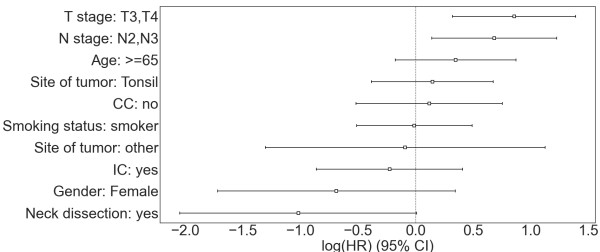

(a) OR plot of Cox model excluding cluster labels as a predictor

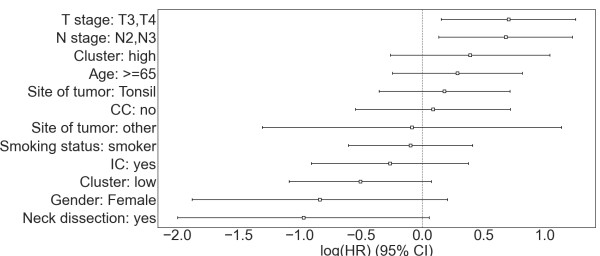

(b) OR plot of Cox model including cluster labels as a predictor

Fig. 7. Odds ratio for the Cox model predictors when (a) the cluster labels are excluded from the model versus (b) when the cluster labels are included as predictors in the model.

## V. DISCUSSION

The Bi-LSTM accurately predicts PRO data. As a longitudinal model, it is able to predict late toxicity (M12) using data available at diagnosis. While symptoms such as nausea, vomiting, and skin have very low RMSEs ($\leq 1$) and symptoms such as dry mouth and taste have larger RMSEs ($< 2.5$), we argue that the performance of Bi-LSTM should be measured on the latter. The reason is that 12 months after the end-of-treatment, dry mouth and taste are two of the long-term toxicities that preserve moderate-to-severe ratings for a large number of patients, whereas very few patients experience moderate or severe nausea or vomiting. Consequently, the Bi-LSTM can achieve a low RMSE for these symptoms by predicting values close to zero, and a relatively higher RMSE by predicting moderate ratings for dry mouth for patients with severe dry mouth. When evaluating the AUC performance of the LSTM for predicting mild-to-severe symptoms, the LSTM achieved an impressive AUC of 81% and 82% for taste and dry mouth at M12.

Moreover, clustering the patients using the predicted PRO data into low, mild, and high symptom burden groups simplifies the integration of PRO data into the predictive models. It is worth noting that the novelty of including a PRO-based cluster, regardless of the time point, improved model performance for both toxicity and survival.

From our results, even the inclusion of clusters from symptom ratings at baseline improves performance. Not surprisingly, the largest performance improvement is seen on toxicity prediction, for xerostomia 12 months after treatment. Xerostomia is one of the most prevalent radiation-induced toxicities and patients often experience it concomitantly with other symptoms. When the logistic regression model does not include the cluster labels, AUC over the test set is 54%, which improves to 64% if baseline clusters are included, and reaches 86% when the clusters using the month 12 predictions are included in the model.

Furthermore, the inclusion of the symptom clusters also improved overall survival prediction. While the inclusion of the clusters from the month 12 predictions improves the test c-index from 62% to 67%, the best-performing model is the symptom clusters over the baseline ratings (test c-index of 69%). This can be attributed to the fact that the Bi-LSTM does not model treatment or survival and will predict symptom ratings regardless of survival outcomes. Lastly, the symptom burden experienced before treatment at baseline can be a proxy for performance status and be predictive of survival early on.

This study is not without limitations. First, since patients missing baseline ratings were imputed using mean imputation,

the long-term predictions for these patients are biased towards the mean values. In the future, it would be worth exploring other imputation techniques that could offer less biased results. Moreover, head and neck cancer patients have experienced improved survival in recent years with 92% survivors in this cohort. The large proportion of censored patients sometimes without enough follow-up makes it hard to obtain a better c-index estimate. Last but not least, as typical of retrospective analyses performed at single institutions, the database used to generate these models consisted of patient data from a single tertiary cancer center and may reflect a patient sample that is not generalizable to the general population. As such the presented results are not yet suitable for general use prior to validation of the predictive models with external datasets. Our future work plans to apply the proposed model to prospective data from the same institution as well as from another institution that can serve as external validation. Even though the collection of PRO data is not standardized across institutions, one advantage of our proposed approach using patient stratification is that it would be easier to generalize to other PRO data.

## VI. Conclusion

The Bi-LSTM model proves to be an effective and powerful network to model PRO data and predict long-term toxicity. Clustering the patients using the predicted PRO data into low, mild, and high symptom burden groups simplifies the integration of PRO data into predictive models. Not surprisingly, the largest performance improvement is seen in toxicity prediction. Furthermore, the inclusion of the symptom clusters also improved overall survival prediction.

In future work, we will evaluate additional long-term toxicities, such as osteoradionecrosis, dysphagia, and sleep disorders, that affect quality of life in HNC patients. We will also examine how PRO-based patient clusters evolve before, during, and after treatment, and how these patterns relate to survival and toxicity. This temporal analysis will explore how early or ongoing symptom burdens impact long-term outcomes, potentially guiding timely interventions to reduce toxicities and improve survival.

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
