# OpenReview forum: "PRO-based Stratification Improves Model Prediction for Toxicity and Survival of Head and Neck Cancer Patients"
_IEEE.org/EMBS/BHI/2024/Conference — IEEE BHI'24_

### Official Review · Reviewer_oc16 · 2024-07-21
**Revision round 1**

**Overall Rating:** 7
**Confidence:** 3

**Review:**

Although the authors strove to present a lot of results about their model in a single paper, as a reviewer, I found it difficult to understand most of the technical details present in the Methods section due to the large amount of performance metrics. Allied to this, the lack of references to other related works in the field makes it difficult to fairly assess the novelty and impact of the current study. Moreover, models based on response to questionnaires usually do not perform as well as other more robust quantitative models and, as the authors correctly pointed out in the Discussion section, the presented results are not yet suitable for general use prior to validation of the predictive models with external datasets.

**Other Quality Metrics:**

(a) Clarity of writing: fair
(b) Clinical significance: fair
(c) Methodological novelty: poor
(d) Experiments and Results: poor

**Questions For The Authors:**

1 - Section II. Methods and Materials. B. Bi-LSTM model: authors mention the computational platform used for training, but not the computational workload or execution times involved in the model.
2 - Misplaced figures 5, 6, and 7 in the middle of the References sections, which contributes to a disorganized manuscript structure.

**Strengths:**

1 - The large amount of data collected (937 HNC patients)

**Summary Of The Paper:**

The present manuscript presents a Bi-LSTM model to predict long-term toxicity in head and neck cancer patients using patient-reported outcomes.

**Weaknesses:**

1 - Absence of a "Related Works" section in the manuscript.
2 - No comparison metrics with other works published in the literature.
3 - Missing baseline ratings (and others) for some patients (incomplete data).

---

### Official Review · Reviewer_H6vi · 2024-07-28
**Summary of Review for PRO-based Stratification in Head and Neck Cancer Prediction Models**

**Overall Rating:** 8
**Confidence:** 5

**Other Quality Metrics:**

Clarity of Writing: Good
Clinical Significance: Great
Methodological Novelty: Excellent
Experiments and Results: Good

**Questions For The Authors:**

n.a

**Strengths:**

As a reviewer, I have found several promising results in the project titled "PRO-based Stratification Improves Model Prediction for Toxicity and Survival of Head and Neck Cancer Patients":

Incorporating patient-reported outcomes (PROs) into predictive models is novel, addressing the limitations of traditional clinical models. The use of bidirectional long-term short-term memory (Bi-LSTM) recurrent neural networks efficiently handles the complexity of long-term symptoms, demonstrating the potential for deep learning in medical research in.

The inclusion of symptom weight groups significantly improves the predictive performance of both logistic regression and Cox regression models, as evidenced by the increased AUC and concordance index The study benefits from a large and comprehensive dataset of 937 patients who they are available for head and neck cancer, which provides a solid foundation for research and model training

The findings have practical implications for personalized treatment planning, and may improve patient outcomes and quality of life. Furthermore, the paper is well organized, with procedures, results, and implications clearly explained, making it easy to follow and understand.

The work represents a significant advance in the use of PRO and advanced modeling techniques to provide treatment-related toxicity and survival in head and neck patients improved prognosis in cancer The new techniques, robust data analysis, and practical implications are particularly noteworthy.

**Summary Of The Paper:**

The paper titled "PRO-based stratification improves model estimates of toxicity and survival of head and neck cancer patients," examines the impact of incorporating patient reported outcomes (PROs) into prognostic models for head and neck cancer patients Primary objective of this study To assess whether classification of patients based on estimated 12-month posttreatment pro provides an accurate prediction of radiation-induced toxicity (mainly xerostomia) and overall survival patients with head and neck cancer have increased or.

The study used data from 937 patients with head and neck cancer treated at MD Anderson Cancer Center from 2010 to 2021. This data includes clinical data, treatment data, and PROs collected over time diversity before, during, and after treatment Imputation of missing data and prediction addresses future symptoms using this model using bidirectional long-term memory (Bi-LSTM) recurrent neural networks use to report symptoms. Hierarchical clustering is applied to Bi-LSTM-predicted symptom ratings by classifying patients into low, medium, and high symptom burden groups and then using these groups as predictors in logistic regression and Cox regression models in. A logistic regression model is used to predict the occurrence of xerostomia at 12 months, and a Cox proportional hazard regression model is used to predict overall survival

Adding symptom weight classes significantly improved the predictive performance of both logistic regression and Cox regression models For xerostomia prediction, the AUC improved from 0.54 to 0.86 by adding symptom weight classes. The concordance index (C-index) improved from 0.62 to 0.67 when groups of traits were added to predict survival. Patients were carefully divided into three groups (low, moderate, high) based on their predicted symptom burden at 12 months post-treatment. These groups correlated well with clinical characteristics and provided important predictive power for both toxicity and survival outcome.

The study highlights the use of PRO data and advanced modeling methods such as Bi-LSTM to predict long-term outcomes in cancer patients. It highlights that incorporating PRO-based stratification into predictive models can significantly increase their accuracy, supporting better individualized treatment planning Research concludes that PRO-based stratification is a valuable resource includes prognostic models for patients with head and neck cancer, improving predictions of radiation-induced toxicity and overall survival This approach enables the development and execution of a more appropriate treatment regimen effectively manage long-term adverse events for patients.

**Weaknesses:**

The study, titled "PRO-based Stratification Improves Model Prediction for Toxicity and Survival of Head and Neck Cancer Patients," has two major weaknesses: First, the reliance on data from a single institution limits the findings all the limits. Second, the lack of external validation with datasets from other organizations raises concerns about the robustness and performance of the predictive models. Addressing these issues will enhance the reliability and clinical significance of the study.

---

### Official Review · Reviewer_Sbxk · 2024-08-12
**better prediction of cancer survival and treatment side effects by including patient-reported outcomes**

**Overall Rating:** 7
**Confidence:** 3

**Other Quality Metrics:**

(a) Clarity of writing; excellent
(b) Clinical Significance; great
(c) Methodological Novelty; (unable to evaluate)
(d) Experiments and Results; great

**Questions For The Authors:**

I do not have any questions

**Strengths:**

This is a straight-forward argued analyses with a strong rationale and strong methods, and a large-enough n.

**Summary Of The Paper:**

The paper reports on the effects of including patient reported outcomes in models predicting cancer survival and side effect burden.
Data from MD Anderson Cancer Center is used, specifically the MD Anderson Symptom Inventory (MDAS) and the analysis is limited to head and neck cancers. Data / medical records from > 900 patients is included.  Two different models are used to predict the two different outcomes. for toxicity (side effects) vs. survival. Results show that including PRO data improves predictioin of survival and toxicity, while the improvement is greater for toxicity.

**Weaknesses:**

none noted

---

### Decision · Program_Chairs · 2024-09-23

Accept